# Using Supra-Covered Bonds to Enhance Liquidity in the Euro Area: Assessment of Advantages for the Banking Sector

**Matteo Salto [1,†], Stefano Zedda [2,\*] and Stefan Zeugner [1,†]**

[1] European Commission, Directorate-General for Economic and Financial Affairs, 1049 Bruxelles, Belgium; Matteo.SALTO@ec.europa.eu (M.S.); Stefan.ZEUGNER@ec.europa.eu (S.Z.)

[2] Department of Business and Economics, University of Cagliari, 09100 Cagliari, Italy

\* Correspondence: szedda@unica.it

† The authors are fully responsible of the opinions expressed in the paper and the opinions here expressed cannot be attributed in any manner to the European Commission.

**Abstract:** The discussion on the necessity of a larger volume of very highly quality liquid assets (VHQLA) in the euro area has been very extensive. The debate on expanding the pool of comparable euro area assets focuses on "safe assets", often on various combinations of government bonds, most of which would not entail a strong increase in euro VHQLA. This paper explores a different option, complementary to the existing ones, based on the creation of a safe European asset backed by fully private assets. The paper proposes the issuance of supra-covered bonds by a central European institution. The latter are bonds issued by the central issuer and backed by covered bonds, which banks would have created using their mortgages as their cover pool. The aim is to increase substantially the outstanding amount of euro VHQLA. Such an asset would also be very beneficial during crisis periods, such as the current COVID19 crisis, by allowing banks to transform mortgages into very high quality liquid assets that can be used for funding and as a collateral in operations with the Eurosystem, thus enhancing the possible credit to sustain small and medium-sized enterprises (SMEs). This paper assesses the main effects of such a proposal on banks under different possible scenarios.

**Keywords:** supra-covered bonds; liquidity; euro

## 1. Introduction

Since the sovereign crisis of 2010, different proposals have been made on how to expand the supply of liquid safe assets in the euro area. While the supply of high quality liquid assets (HQLA) is relatively large, in 2019, the pool of very high quality (rated AA or better) and highly liquid assets (VHQLA) was only half of it and amounted to above EUR 5 trn, mostly of highly rated government and agency bonds (see Grandia et al. 2019, for a discussion). To compare, federally-backed US debt securities alone amounted to broadly USD 25 trn.

Existing proposals to enlarge the pool of euro liquid safe assets so far focus on enlarging the supply of government bonds, the safe asset par excellence, or combining existing government bonds for the provision of such assets. There are different arguments in support of an increased supply of safe assets, which is often seen as a complement to the euro area architecture. The first type of argument relates to monetary policy. An increased supply (if appropriately distributed across the euro area) would allow a symmetric transmission of monetary policy impulses to all firms, independently from their location and the quality of their governments, and facilitate monetary policy operations. A second group of arguments relates to the possibility of enhancing the use of the euro as a reserve currency and more in general in the international financial markets and in invoicing practices

Thus, the debate mainly focused on the possible issuance of joint and common debt by euro area governments. Some relevant examples are the Blue Bond proposal (Delpla and von Weizsäcker 2010) and the European Safe Bonds (Brunnermeier et al. 2017), which raised the interest of the European institutions, including the pledge to present "with a 2025 perspective ( . . . ) the possible development of a euro area safe asset" (Juncker and Timmermans 2017). Zettelmeyer and Leandro (2018) provide an evaluation of four main approaches to create "safe assets" or asset portfolios for the euro area. A recent literature review of existing proposals is presented in Giudice et al. (2019).

There are broadly two types of difficulties related to the creation of a centralized liquid and highly rated asset in the Euro Area. A first group of difficulties relates to the diverging views across euro area members, with a group of member states pointing to the benefits of risk sharing in government debt and another group of countries stressing the risk of an increase in the overall debt due to moral hazard considerations. This debate is also reflected in the ongoing debate on the regulatory treatment of sovereign exposure, a politically divisive discussion in the context of the completion of the banking union. The second objection to the creation of a centralized liquid and highly rated asset in the Euro Area refers to the fact that monetary policy has been conducted effectively without such a common asset. As indicated above, the pool of HQLA in the euro area is relatively large and the European Central Bank (ECB 2008) has been effective in adapting collateral rules as needed. For example, in April 2020 during the COVID19-induced crisis, the ECB launched new liquidity operations and, few days later, revised temporarily its collateral framework in order to secure that banks could effectively pass on this liquidity to the real sector, thereby smoothing the transmission of monetary policy to the entire euro area economy.

This paper explores a complementary option, namely the creation of the supply of a very high quality and liquid European assets based on the intermediation of fully private assets. The type of safe asset that is needed to allow a smooth monetary transmission or to enhance the use of the euro in international markets does not need to be necessarily issued by governments nor be based on such issuances (such as the ESBies proposal for example). The proposal has the advantage of avoiding the main political problems behind the proposals of increasing safe (government) assets. Moreover, the present proposal is complementary to the "safe asset" literature and to existing "safe assets" proposals. First, because of any absent agreed risk sharing on government debt, constructions based on existing euro area government bonds would realistically increase the pool of VHQLA compared to the status quo only in a marginal manner. Second, the proposal would give more operational instruments to the banking system and achieve a more uniform monetary policy transmission across the euro area under the assumption of a relatively uniform distribution of such assets across the euro area.

The proposal revolves on the transformation of covered assets by banks into a pan-European AAA-rated asset. Such an asset would be particularly beneficial during crises, such as the current COVID19 crisis, by allowing banks to transform mortgages into VHQLA that can be used for financing and as a collateral in operations with the Eurosystem. On the one hand, if used by banks in all euro area countries, such an asset would support monetary policy by allowing better transmission of monetary policy. On the other hand, it would support banks with funding and liquidity in case of the emergence of tensions in the sovereign markets, especially if those banks are located in "weak" countries.

## 2. The Proposal: A Supra-Covered Bond

The idea proposed in this paper is based on the observation that the euro area currently accounts for more than EUR 5 trn of outstanding mortgages, part of which are issued by small banks or have peculiar characteristics so are not liquid enough to allow for the formation of a single-issuer benchmark curve.

Those mortgages could be used to construct a (potentially large) pool of euro-denominated AAA-rated supra-covered bonds, issued by a central European institution, to be exchanged with the covered bonds. Banks would offer the latter to the central institution, backed by their mortgages. The primary aim of the paper is to assess the main effects for banks of the proposal.

In short, a central issuer would buy covered bonds by individual banks, based on mortgages as cover pool, and fund them through large, high quality and liquid bond issuances. More precisely:

1.　Banks bundle mortgages into covered bonds equivalent to investment grade. Note that covered bonds leave junior mortgage risk with the bank; thus, covered bonds are typically rated as highly as sovereigns.[1]

2.　Individual commercial banks issue covered bonds directly to a central European issuer, according to a standardized procedure, thus being able to issue small-size bonds and saving part of the usual issuance costs.

3.　To fund those assets, the central issuer issues bonds, which normally are AA+ or AAA-rated.

To the extent that a large number of banks finds it convenient to transform their mortgages into such a liquid AAA asset, the issuer of such bonds could become the largest benchmark issuer of liquid and high-quality debt in the euro area. Diversification across housing markets and issuers allows for achieving a AAA credit rating. It would be liquid and high quality as the issuer runs minimal default risk and overhead cost by holding covered bonds, rather than mortgages directly.[2]

Given the amount of assets potentially offered by the central issuer and the fact that the supplier of the assets is unique, the asset is potentially very liquid and has very large volumes. As the central issuer would have covered bonds on the asset side and the risk diversification involved, plus the guarantee of the issuing bank, there is not much doubt that the asset will be high quality.

For investors, this model would emulate US mortgage-backed securities, though with key differences on the banking side: covered bonds are standardized and are, since 2019, governed by the European covered bond Directive. This allows circumventing due diligence costs, and country specific legal issues that would affect the outright purchase of mortgages. Critically, in contrast to the US model, covered bonds do not involve securitization and leave mortgages on the balance sheets of banks, which have, therefore, an incentive to maintain the quality of the pool of mortgages.

In operational terms, we do not discuss here the ownership structure and the governance of the central issuer, however important, because first we want to make the case that such a construction would be a relevant addition to the current state of affairs, in particular for banks. However, one could imagine that such body could be affiliated with, or modelled after, the European Investment Bank (EIB), with specific arrangements for the absorption of banks' covered bonds. Two important issues related to the set-up of this body are also not covered here.

The first concerns who bears the risk of this new body. The normal business risk is in principle covered by the amount of capital required to the central issuer. In this proposal, the risk coming from mortgage bundles stays within the issuer bank, unless, as discussed below, the design of the central institution is such that part of this risk is transferred from banks to the central issuer (below we suggest a possible transfer of the order of 5 to 10%). At first look, this implies that the central issuer of supra-covered bonds would need a small capitalization, due to both the low riskiness of its assets (namely the covered bonds) and to the large diversification of the underlying risks.[3] The appropriate capitalization level of the issuer can be estimated by means of the current regulation on the basis of the riskiness of the actual mortgages and their duration[4]. This capital coverage can be mitigated not only by the benefits from the vast diversification across countries of the underlying real estate

---

1　In contrast to asset-backed securities, covered bonds do not bundle mortgages off a banks' balance sheet. Mortgages are kept on the bank's balance sheet and used as collateral to cover a senior bond with a substantial safety margin.

2　In contrast to bundled mortgages, covered bonds are a market security that is subject to harmonized rules and oversight since Directive (EU) 2019/2162) of November 2019 harmonised certain aspects of the conditions for the issue of covered bonds. Pricing and gauging default risk can rely on standard models. Possibly other aspects may need to be harmonized.

3　On top of the differentiation from the presence of different banks and households, the cycles of the house prices are very idiosyncratic across the euro area.

4　In case some risk was transferred to the central issuer's balance sheet as proposed later, one should also take into account the fraction of risk transferred

risk, as normally accepted in finance practice, but also by the diversification of the portfolio of banks. This holds in particular for banks of medium and small size, whose portfolios tend to have a larger share of loans. A test in Zedda et al. (2018) confirms that the higher the loans incidence on total assets, the lower the correlation on banks' results, thus reducing systemic risks. As indicated below, it is possible to add to this scheme the transfer of a small fraction of the risk (for example of the order of 5% to 10%) to the central issuer, who benefits from high diversification. In this case, clearly the central issuer bears the risk of part of a local crisis, involving just one or a limited number of small banks. This risk can be covered via the pricing system and via an increased capital of the central issuer.

However, while the existence of supra-covered bonds can limit contagion risks due to the reduced correlation with sovereign bonds and public finances, a decision would have to be taken on how to cover losses in case of a crisis of systemic euro area importance. In this respect, an interesting development to be analyzed, is whether the cover pool could also be made of loans to small and medium-sized enterprises (SMEs). The latter would support small banks and would be justified by the fact that lending to SMEs is very differentiated across banks by country and sectors, as confirmed by the recent literature on systemic risks. In particular, Bams et al. (2015) document how SME lending tends to increase the risk diversification of portfolios across banks, resulting in a lower correlation of the risk of these portfolios with common variables such as GDP at the national or euro area level. Even if this would not exclude some impact of regional crisis, the regional concentration of lending within each small bank would be more than compensated by the diversification across banks. Due to the diversification, each bank just represents a small share of the central issuer assets portfolio, but this does not exclude the possibility of losses possibly affecting the central issuer in case of a systemic crisis, leading a large share of the mortgage customers and of the issuing bank in the euro area to default. As said above, the present paper leaves it open how to face a systemic euro area crisis, which would likely require massive interventions from policymakers[5]. However, it should be noted that the present proposal does not increase the risk of such a systemic crisis per se. Indeed, individual banks continue facing the risk that they generate with their loans, contrary to the US experience, where banks could transfer such risk on the centralized institutions. The principle that banks cannot transfer risk on the central issuer even if would not cover any possible crisis, certainly reduces the related risks.

The second issue is the price at which the exchange would take place between the covered bonds issued by banks and the supra covered bonds issued by the central issuer. Pricing is an important aspect of the construction in that it has to allow the central issuer to be able to issue AAA-rated bonds. This result should be secured without overburdening the banks, ideally using standardized criteria. A simple model for pricing by the central issuer of covered banks bonds could be based on the Grothe and Zeyer (2020) proposal of a unified risk metrics. In their paper, the authors analyzed the risk characteristics of covered bonds showing that there is a significant heterogeneity of covered bonds, which they group into issuer risk, overcollateralization, cover pool risk, mismatches between covered bond and cover pool cash flows and maturity risk.[6] Other differences could be relevant for developing a standardized pricing practice, such as differences in national legislation and practice on the right to seize the property of defaulting borrowers, which affect the loss given default (LGD) estimation. The decision about pricing is part of the more general issue of the creation of an appropriate, market-based governance structure, which goes beyond the scope of the present paper.

The third issue, partially related to the previous ones, relates to the regulatory treatment of the bonds emitted and the legislation that applies to the central issuer, in particular, whether the central issuer has to be treated like a fully private company and its bonds like normal bonds issued by a

---

[5]  There can indeed be a systemic crisis that involves the mortgage markets of many European countries at the same time, even if this is an extreme scenario, which case would require an external intervention, and possibly the issuance of a government safe asset, which is not the subject of this paper.

[6]  It should be noted that the issuer risk and the cover pool risk reflect the sovereign risk of the country in which the bank is located. This is already reflected in ratings of covered bonds as shown by Grothe and Zeyer (2020).

private actor for example with respect to the implementation of the Basel II criteria or whether it has some kind of public nature.

The present paper briefly presents on the main advantages of the proposal for commercial banks. One could wonder whether banks would in practice use such a scheme when in need for liquidity. The suggested answer is positive, and this conclusion is supported by already existing practice.

For example, BPER banca, an Italian banking group included in the EBA 2019 EU-wide transparency exercise, already in 2009 carried out, via a Special Purpose Vehicle, "a securitisation of performing residential mortgages" … with a view to strengthening the funding available to tackle liquidity risks".[7] The operation, worth EUR 1.9 bn and not meant to " transfer to third parties, with respect to the originator bank, the real credit risk associated with the underlying loans", produced a first tranche of senior AAA rated securities (91% of the total amount), a second tranche A rated (2% of total) and a third non indexed tranche (4% of total). All securities were absorbed by BPER Banca: "the objective of this operation, not involving the market, was to create a reserve of liquidity via the issue of securities eligible for refinancing with the ECB and for use as a guarantee for other funding transactions"[8]. A similar operation was carried out in 2017 by Banco di Sardegna, a BPER subsidiary, for a total amount of around EUR 1.5 bn, producing a Aa2-rated senior tranche of 76% of total, and a junior unrated tranche for the residual 24%[9].

More in general, Koulischer and Roy (2019) show that in 2015 around 30% of the outstanding bank loans to firms, households and governments are already used as collateral for funding or liquidity purposes. Of these, around two thirds are used for funding purposes and the rest is used for liquidity purposes, by pledging them directly or after pooling them into asset-backed securities or covered bonds. The authors estimate that around 700 billion in loans in 2015 backed covered bonds or asset-backed securities, of which roughly 400 billion corresponded to mortgages backing covered bonds or asset-backed securities.[10] Grothe and Zeyer (2020) report that "there is a significant heterogeneity of covered bond programmes with respect to various dimensions of their risk characteristics" and that "the credit ratings of these instruments tend to be relatively high though, with 101 out of 198 bonds rated AAA".

The high rating is expected when the issuances are aimed at transforming illiquid mortgages in high quality collateral for covering liquidity needs, and this is also supported by the level of overcollateralization, which is reported by same paper to be of 72% on average.

Therefore, banks having access to the ECB liquidity operations have long been creating securities from mortgages to be used as collateral in refinancing operations "own-used" covered bonds in legal jargon. However, the existence of the supra-covered bond would bring advantages to commercial banks mainly relating to funding costs:

1.  *Liquidity premium:* By bundling assets, the central issuer should reap a significant liquidity risk premium compared to the price the underlying covered bonds would achieve on the market. The issuer would share a part of this premium with originating banks. This provides an advantage in terms of both funding and liquidity.
2.  *Scaling:* By submitting standardized covered bonds to the central issuer, banks can fund mortgages in considerably smaller chunks than at present. This reduces the liquidity premium that normally affects the issuance of small-scale bonds. In particular, small banks could benefit from funding advantages similar to larger banks.

---

[7]  See BPER annual report 2017, page 395.
[8]  Same as previous note.
[9]  Same as previous note, page 396.
[10] According to the authors roughly a quarter of total loans are extended to firms, a third is extended to households and a tenth to government entities, and around 55% of ABS are backed by mortgages. However, in 2007 the proportion in the cover pool between private and government loans was roughly 40% and 60% respectively, see ECB (2008).

3.  *Duration risk sharing:* The variety of mortgage durations across the euro area allows the central issuer to offer banks low-cost funding of the duration profile they require.
4.  *Issuing costs:* Issuing standardized covered bonds to a single buyer should result into issuing costs significantly below those of issuing a "normal" covered or bank bond.
5.  *An enlarged possibility to improve the liquidity coverage ratio.* The purchase of the centrally issued supra-covered bonds would allow banks to have additional high-quality assets to be used in monetary policy transactions and to diversify their exposure from government bonds. Under certain conditions, the latter could improve the liquidity coverage ratio (LCR) without the use of government bonds. In case some risk is transferred to the central issuer, the proposal would imply a diminution of capital requirements.

The advantages above can be enjoyed by all banks, even if in different degrees. The proposal can improve the level playing field across banks of different size: indeed, until now, these kinds of operation are only feasible for large banks, while small banks have to find other, more costly, funding strategies. The proposed centralized issuance of supra-covered bonds would result in a direct access to liquid assets for smaller banks, and in a standardized, more efficient "conversion" of mortgage loans into safe assets for larger banks; the latter would receive actually marketable assets instead of the securities in the example quoted above, formally highly rated but not actually tested by the market.

## 3. Benefits for Banks from Increased Liquidity

Access to liquid assets brings advantages to banks in terms of funding and in terms of access to liquidity in the context of monetary policy operations and regulation. First, in terms of regulation, high asset liquidity supports financial stability. The demand for safe assets depends on the prominent role that this asset class plays in the everyday operations of international financial markets (Golec and Perotti 2017 for a review). To this purpose, the Basel III framework introduced the Liquidity Coverage Ratio (LCR), which measures to which extent banks' liabilities volatility is covered by high-quality liquid assets. More specifically, at least three-fifths of the liquidity coverage requirements have to be met with "level-1" assets, which in the Basel framework are only exposures to governments and other public-sector entities. The LCR, introduced in 2015 in the European Union under the Capital Requirement Regulation, reached its steady-state calibration in 2019. Depending on the regulatory treatment of the bonds issued by the central issuers, banks could use the supra-covered bonds also to fulfil the LCR.

Second, in economic terms, the proposal would give the possibility to banks of generating very high-quality liquid assets. Those can be used for funding, or as collateral in money markets or central bank operations to receive liquidity. This can be useful in crises such as the current COVID19 crisis, not only for banks, but also from a systemic point of view, in that it would facilitate monetary policy action, thanks to the larger presence of collateral in the system.

Based on Christopoulos (2017) one can propose a ballpark figure for the advantage for commercial banks from access to increased liquid funding from mortgages in a stressed period of the order of 200 basis points. Christopoulos (2017) identifies the liquidity risk premia embedded within commercial mortgage backed securities (CMBS) in the US. The paper shows that the share of liquidity premia varies across time and across asset quality. In particular, the entire spread of BBB securities with respect to government bonds is due to credit risk, while the liquidity premium explains up to 90% of the same spread of AAA securities, with very large variations across periods. In particular, regular liquidity spread was high during the financial crisis, but disappears when the Federal Reserve started its non-standard monetary policy. The difference in liquidity spread gives the estimate of around 200 basis points at the peak of the financial crisis provided above.

The previous estimate provides a likely upper bound to the positive liquidity effects for banks from the introduction of supra-covered bonds, as those effects are based on price effects derived from the National Survey of Mortgage Originations (NSMO) under circumstances in which liquidity was badly

needed. However, those findings allow for some rough evaluations on the effects of the supra-covered bonds (under the assumption that the supra-covered bonds can be used in the banks' operations).

First, in order to have a figure for the potential total advantage for banks, it is necessary to assess the volumes of the new AAA-rated asset that banks could potentially generate—if they find it convenient. With reference to the expected liquidity volume coming from supra covered bonds for banks, in aggregate it seems realistic to hypothesize that 1/3 of the over 5 trn of mortgage loans present in the euro area can be converted into a volume of 2 trn euro of supra-covered bonds. This is at least a large as the 1.5 trillion of covered bank bonds in the universe eligible for collateral in Q1 2020 according to the ECB. Banks have already obtained funding from the Eurosystem selling their covered bonds in the context of the Covered Bonds Purchase Programs. The current holdings of covered bonds by the Eurosystem under the Covered Bonds Purchase Program 3 are of the order of 270 million in April 2020.

On top of funding, banks can use the covered bonds to also obtain liquidity in a large systemic crisis, even if facing such a type of crisis is not the main aim of the scheme proposed in this paper. During crises, the financial system typically experiences a "flight to quality". For example, in the 2008 global financial crisis the interbank market significantly reduced its lending volumes. Since then, a significant share of the interbank loans is collateralized by sovereigns (of safe countries) or by VHQLA. This is in line with the literature. For example, Reinhart and Rogoff (2009), show that in times of systemic crisis, sovereign debts tend to increase while their credit standing tends to diminish. This dynamic can generate a shortage of safe assets. In similar cases, the availability of a higher volume of VHQLA can mitigate the safe assets shortage thus reducing the liquidity premium. Indeed, during the COVID19 crisis, as indicated above, the ECB has increased its liquidity provision and has made sure that banks have sufficient collateral to pledge in order to obtain that liquidity. In these terms, liquidity also has a cost for banks when interest rates are null. Such a cost depends on the general economic situation and on the situation of the bank.

In the euro area, banks have pledged a total value of 408 bn covered bonds, after valuation and haircuts. Cost for liquidity in money markets can be assessed by looking at prices for general collateral.

There are no available estimates of the effect on liquidity by the possibility of pledging covered bonds. However, an indication of the order of magnitude of the benefits for the euro area banking system when the emergency related to the COVID19 abates, can be computed by imagining that euro area banks would have had to find the necessary liquidity to pay back their 2019 (targeted) long-term refinancing operations (TLTRO) loans in 2020 as originally scheduled. Such a reduction in yields could have brought to the banking system of the euro area, savings of the order of 50 bn over the next two years: liquidity from long-term monetary policy operations to banks was worth around EUR 727 billion and was expected to drop to around EUR 376 billion by 2020. Depending on the market price of liquidity, banks could have realized sizable savings.

## 4. Effects on Banks' Balance Sheets When Liquidity Price Is Low

In the current situation, banks have an easy access to short-term low-cost liquidity and have had the opportunity to access medium-term liquidity using the TLTROs proposed by the ECB, so that liquidity needs are reduced.

However, even for banks that already have direct access to the ECB liquidity provision, supra covered bonds would offer some advantages.

Firstly, direct liquidity savings could accrue to banks in the context of the liquidity operations. It seems reasonable that the assets received from the central issuer would face a more advantageous haircut than current covered bonds, and that the central issuer would be reflecting its pooling advantage into the price to banks. Therefore, the value for the bank from this operation should be positive.

Second, there are advantages related to the standardization of the evaluation of the covered bonds, which is required in order for the central issuer to set transparent prices.

Finally, it is possible in principle to add to the scheme a (small) risk-transfer from banks to the central issuer, without compromising its AAA rating, with a minor increase in capital. Obviously,

in this case banks have a clear additional incentive to adhere to the scheme from the reduction in risk weighted assets (RWA), as the scheme would contain a partial risk transfer to the central issuer. For a given risk transfer share (tentatively, 5–10%), the individual commercial bank benefits from a reduction in the minimum capital requirement, by an amount nearly equal to the share of risk transferred to the central issuer. This is a key benefit for a large share of banks. In this case, the individual bank obtains supra-covered bonds for a value similar to that of the bonds transferred to the issuer, but with the highest rating and no risk weighting, as it happens for sovereign bonds.

The presence of such risk-sharing top up would add a new euro-area policy instrument, as the effect on banks' RWA can be highly significant in crisis situations, such as the present COVID19 crisis, in which the loans' probability to default of individual banks increases, rising the correspondent risk weighting and the minimum capital requirements. The latter increase prevents banks from fully transmitting monetary impulses to the economy. Indeed, the required increase in banks' capital, can potentially act pro-cyclically inducing a reduction in credit. Such credit reduction is typically more affecting SMEs due to their reduced possibility of a direct access to financial markets (Bussoli and Marino 2018).

In short, even in situations in which liquidity collection is not an issue, banks will anyway benefit of a liquidity coverage for the amount of the obtained supra-covered bond. The latter can be used for backing interbank liabilities and more mortgage loan issuing, in so providing a potential for credit volumes to increase. Providing additional liquidity to small banks, which do not have direct access to the ECB balance sheet, is one of the objectives of this proposal. This will certainly go in the direction of increasing the loans volume. This can possibly have adverse consequences on the loan quality, given the high degree of competition in this market segment, in particular in periods of booms.

This concern is, however, mitigated compared to the period before the financial crisis. Banks issue new loans of lower quality (and higher risk) only if the increased loan provisions, comprising the capital absorbance and its related costs, brings higher revenue. Capital requirements have been strongly reinforced since the crisis, by means of the Capital Requirement Directive IV and its subsequent amending. Moreover, new macro/microprudential powers were given to regulators or central banks, that introduced a serious monitoring and curing of those risks so that at least the pro-cyclical effects of the increased loan provision should be limited.

More, the standardization of the issuance will substantially lower costs even for the average to large size banks, which already have a possibility to issue marketable "own used" covered bond, but at the cost of the whole issuance process, including a double rating.

This framework can be even more interesting for small banks, which typically invest a large share of their portfolio in mortgages and SME loans and do not have direct access to central bank liquidity operations. Those banks could have a new direct access to liquidity, in that the newly created liquid assets provide them a large—compared to their assets—amount of new collateral.

## 5. Relevance of Private-Sector-Based High-Quality Assets

In case sovereigns of one country lose their investment grade, the pressure faced for the subscription and holding of the sovereign bonds is coupled with the non-investment grade of its rating. This would imply that banks located in a non-investment grade country need to own other AAA-rated assets both for liquidity needs and for backing interbank exposures. Thus, the availability of high-quality liquid assets coming from the transformation of mortgage loans into supra-covered bonds can be very important in this scenario. Moreover, the non-direct linkage between sovereign bonds and mortgage loans on one side, and supra-covered bonds on the other side, can have important effects in reducing the doom-loop between sovereign and banks.[11]

---

[11] It should be noted, however, that such risk cannot be completely eliminated. Thus, the paper does not assess the situation in which a real estate crisis and sovereign bond crisis happen. In this case, temporary losses for the central institution are very likely to emerge. We think the central issuer should be accredited to the ECB for liquidity assistance, but this is not discussed in the present paper.

The advantages for the banks of the distressed country are of two types. First, banks have access to cheap collateral. The amount of collateral available is equal to a share of the total of mortgages (see above), and for large banks such collateral would secure the access to the overnight market. It is indeed possible that the ECB does not intervene as during the crisis, as far as only one country is involved in the crisis, inflation remains around target and monetary transmission is not disrupted in the rest of the euro area. This advantage cannot be monetized. Second, if government yields loose a substantial amount of their value, so that it becomes very expensive to use one country's sovereigns as collateral, banks could partly substitute the sovereigns with the supra-covered bonds for this purpose thus mitigating the negative spillover from the sovereigns.

## 6. Conclusions

This paper proposed the creation of a central institution issuing supra-covered bonds in exchange of the covered bonds that banks would offer based on their mortgages. This proposal can have important effects on the stability of the euro area. Firstly, it can enlarge the supply of highly liquid and AAA-rated assets in euros, which is now relevantly smaller than the corresponding assets denominated in US dollars. Second, it can have important effects on banking activity. On the one side it would provide a simple access to AAA-rated and highly liquid assets, which can be used for backing interbank loans. On the other side it would reduce the risks in case of one country sovereign bonds lose the investment grade rating. This framework can be even more interesting for small banks, which typically invest large shares of their portfolios in mortgages and SME loans and do not have direct access to central bank liquidity operations.

**Author Contributions:** Conceptualization, S.Z. (Stefan Zeugner), M.S. and S.Z. (Stefano Zedda); writing—original draft preparation, S.Z. (Stefan Zeugner) and M.S.; writing—review and editing, M.S. and S.Z. (Stefano Zedda). All authors have read and agreed to the published version of the manuscript.

**Funding:** This research received no external funding.

**Conflicts of Interest:** The authors declare no conflict of interest.

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
