# Peer review of "Using Supra-Covered Bonds to Enhance Liquidity in the Euro Area: Assessment of Advantages for the Banking Sector"

_jrfm, doi:10.3390/jrfm13120293_

Round 1
Reviewer 1 Report
Referee report on
“Using supra-covered bonds to enhance liquidity in 3 the euro area: assessment of advantages for the banking sector”
submitted to the Journal of Risk and Financial Management
The paper presents an interesting proposal of the pan-European issuance of private sector very high quality and liquidity assets (VHQLA). Given the current scarcity of this category of assets, the proposal has the merit to contribute to the ongoing discussion on the issue. It also relates to the broader and at the same time controversial discussion related to the completion of the capital market union within the Eurozone.
Although the proposal deserves attention, there still remain important issues to be clarified.
- In the introduction, the authors consider the difficulties related to the implementation of the project. In this context, they mention two kinds of counter-arguments: the political opposition and the one related to the effectiveness of monetary policy without the need to issue a common asset. The authors interpret the latter issue as being the main difficulty. However, it seems that the lack of the political will is of even higher importance. This has very much to do with the aforementioned broader discussion on the completion of the capital market union and the extensive fears regarding the issue of mutualisation of risks within the Eurozone. It would be worth exploring this argument in more detail, by showing the reasons for the political disagreement.
- The authors should focus more on the issue of liquidity in general and of the VHQLA in particular as well as on the related risk. From the practitioner point of view, assets are liquid as long as the investors are willing to exchange them. This is customarily not the case under the crisis circumstances, where even very highly liquid assets might face difficulties. The best example confirming that such a sudden turn in the liquidity perception may well happen is offered by the past experiences of the Great Financial Crisis. In turn, provided that the liquidity crisis turns into a solvency crisis, how would the losses be distributed between the involved parts?
- As it is rightly acknowledged in Section 2 of the paper, the model of the pan-European issuance of VHQLA resembles US mortgage-backed securities, which eventually led into the Great Financial Crisis. The authors stress the crucial differences with respect to the US model, being that covered bonds do not involve securitization but rather leave mortgages on the banks’ balance sheets. Whereas this intuitively suggests that to some degree the risk would be diminished – as distributed differently – it is, however, not clear, how the pan-European model makes sure that the past experiences would not repeat.
- An improved access to liquid assets for funding purposes is an indisputable advantage for banks, especially of smaller size. However, it also bears some new risks, as an improved access to liquidity would translate into an increased loans provision by such smaller banks. Given the high degree of competition in this market segment, this could have adverse consequences on the loan quality. It would be necessary to discuss such (indirect) risks more in depth.
- The paper would benefit from a throughout editing work, including the shortening of the sentences. Moreover, the numbering of sections has to be adjusted.
Reviewer 2 Report
I gladly reviewed the paper "Using supra-covered bonds to enhance liquidity in the euro area: assessment of advantages for the banking sector "written by Zeugner S., Salto M. and Zedda S.
The paper proposes the issue by a European institution of a European bond backed by covered bonds created by the banking system using their mortgages as cover pool.
The paper is well written, and properly contextualized in the literature and legislation. The main originality of the paper consists in the fact that this "safe asset" would be based on private assets.
I believe that the contribution is already publishable in this form. My advice is to try to integrate the introduction by further emphasizing the importance of a safe asset within a modern financial system for purposes different from monetary policy.
The demand for safe assets depends on the prominent role that this asset class plays in the everyday operations of international financial markets (Golec and Perotti 2017 for a review). Importantly, safe assets are used by banks and various financial institutions as high-quality (high-liquidity) collaterals in transactions. Recently, new banking regulations have boosted demand for safe assets for prudential purposes. Moreover, the core business of pension funds and insurance companies is characterized by a regular outflow of financial resources which justifies a structural allocation of safe assets in their portfolios (due to the expected stability of their long-term positive yield).
Furthermore, the supply of public safe assets is constrained by a country fiscal capacity and creditworthiness. Reinhart and Rogoff (2009) argue that in times of crisis sovereign debts tend to increase while their credit standing tends to diminish. This dynamic can generate a shortage of safe assets. Gorton et al. (2012) show that the demand for safe assets for the American economy since 1952 has remained constant. This implies that if the availability of public safe assets varies according to the conditions of the economic cycle, the gap between supply and demand has to be filled by private safe assets (Krishnamurthy and Vissing-Jorgensen 2015).
A more philosophical issue is whether a safe asset should be that fixed point capable of being a reference in the pricing of assets. In this case, the safe asset should be created by a public authority outside the market. But this is the theme for a special issue of the journal, not a comment on the paper.
My opinion is that the authors' proposal can be very effective, especially during this pandemic.
Round 2
Reviewer 1 Report
I have now looked at the paper and the corrections made. The new version is much improved, so it would be fine with me to accept the paper.